# Design and Analyses of a Transdermal Drug Delivery Device (TD^3^) [note 1]

**DOI:** 10.3390/s19235090

**Published:** 2019-11-21

**Authors:** Jennifer García, Ismael Ríos, Faruk Fonthal Rico

**Affiliations:** Advanced Materials for Micro and Nanotechnology—IMAMNT Group, Automatics and Electronics Department, Universidad Autónoma de Occidente, Cali 760030, Colombia

**Keywords:** transdermal drug delivery, micro-electro-mechanical systems (MEMS), finite element analysis, microstructures, computational fluid dynamic

## Abstract

In this paper, we introduce a novel type of transdermal drug delivery device (TD^3^) with a micro-electro-mechanical system (MEMS) design using computer-aided design (CAD) techniques as well as computational fluid dynamics (CFD) simulations regarding the fluid interaction inside the device during the actuation process. For the actuation principles of the chamber and microvalve, both thermopneumatic and piezoelectric principles are employed respectively, originating that the design perfectly integrates those principles through two different components, such as a micropump with integrated microvalves and a microneedle array. The TD^3^ has shown to be capable of delivering a volumetric flow of 2.92 × 10^−5^ cm^3^/s with a 6.6 Hz membrane stroke frequency. The device only needs 116 Pa to complete the suction process and 2560 Pa to complete the discharge process. A 38-microneedle array with 450 µm in length fulfills the function of permeating skin, allowing that the fluid reaches the desired destination and avoiding any possible pain during the insertion.

## 1. Introduction

The usage of transdermal drug delivery devices (TD^3^) based on micro-electro-mechanical systems (MEMS) and NEMS technology is increasing [1,2,3]. The transdermal drug delivery (TDD) method refers to medicine administration through the skin at determined depths [3,4,5]. These new devices are facing some challenges, such as the drugs dose effectiveness and the pain caused during the dosing process by the insertion devices. Another approach is dosing from a transdermal patch, which could provide the medicine continuously using the capillarity principle [6]. However, the extraordinary barrier properties of the skins outer layer, stratum corneum, almost wholly block the transport of insulin and other large therapeutic molecules. Chemical, electrical, ultrasonic, and other methods have had some success in increasing transdermal insulin delivery [7,8]. The microneedle array designs differ in needle structure, shape array density, and materials, and these are one of the most modern techniques for drug delivery [5]. A microneedle can be classified on the fabrication process in-plane, which is hard to fabricate with two-dimensional geometry, and out-plane microneedles, which are appropriate for fabrication in two-dimensional arrays by wafer-level processing [9]. An actuated driven membrane is the most common method to make a fluid move in the micro-electro-mechanical world [10]. The main issue is how the membrane gets the mechanical energy to transmit it to the fluid. There are several ways involving electrostatic, magnetostatic, pneumatic, piezoelectric, and many other actuation principles to achieve this [11]. In this paper, we present the TD^3^ design device using computer-aided design (CAD) techniques and the proposed validation design using computational fluid dynamics (CFD) setting simulations that will evaluate the performance of the device. We emphasize the fact that the structural design is oriented to the device functionality and to the way of what an actuation principle refers to, giving us the confidence and ensuring us to have a structural design developed exclusively for the operating conditions and settings that only a TD^3^ will confront.

## 2. Transdermal Drug Delivery Device (TD^3^)

The TD^3^, which has a 38-microneedle array attached, is designed to integrate a thermopneumatically driven membrane that impulses the fluid through them and pyramid-shaped piezoelectrically driven microvalves integrated to correct the flow direction [4,7]. A previous study was presented on the design of the microneedle to understand the best structure that allows delivering the necessary dosage for a patient, without damaging the insulin molecule or the patient feeling pain [12]. The submitted contribution of the microsystem design is in accord with the results of several researchers [3,11,13], which will allow full integration in the future in the third generation transdermal drug delivery [3].

### 2.1. TD^3^ Actuation Principle

The thermopneumatic actuation principle has many coupled physics models, such as solid mechanics, fluid mechanics, and thermodynamics [9,11]. The actuation principle, shown in Figure 1, starts with the gas expansion enclosed in the upper chamber, which is generated by a change in temperature ΔT. Due to the forces generated under the increased pressure in the heated gas, a membrane expands and causes a change of volume and pressure in the lower chamber, ΔV and ΔP, respectively. Valves are strategically positioned to avoid unwanted reversed flows. These valves are known as check valves and in this case, are piezoelectrically driven check valves that operate in synchrony with the actuated membrane.

In recent years, several researchers have worked on the development of piezoelectric (PZT) actuators. In 2016, Dr. Shoji, in Japan, reported a study showing the development of a PZT actuator based on Nafion-117, wherewith a signal square-wave polarization waveform of ±2.0 V, for 2 s, where the fabricated micropump effectively transported water at ~5 L/s [11]. In the same year, research from the Department of Mechanical and Materials Engineering, Queen’s University from Canada, reported the development of a PZT actuator based on poly(dimethylsiloxane) (PDMS), with a peak flow rate of 135 µL/min and a maximum back pressure of 25 mm H_2_O at an actuation frequency of 12 Hz [13]. Furthermore, in India, researchers reported another study of PZT PDMS with pressure-flow characteristics of the micropump of maximum back pressure 220 Pa at a maximum flow rate of 20 µL/min [14].

Once the fluid has left the lower chamber, it flows directly to a set of microchannels that will conduct the fluid to the microneedle array. The suction and discharge are the micropumps basic movements, which will be the main design parameters to take into account during the structural allocation.

### 2.2. TD^3^ Structural Design

The volumes of both the upper and lower chambers are directly responsible for the excellent micropump performance. This is the reason why these chambers were sized first, in order to assign a spatial location that complies with the design requirements to conform to the device [9,10].

#### 2.2.1. Suction Oriented Structural Design

The inlet, or suction, section of the micropump shown in Figure 1, is composed of four inlets and each one of those has a piezoelectrically driven pyramid-shaped microvalve for flow direction control purposes before entering the lower chamber. The total inlet area is 4 × 10^−2^ mm^2^, which does not decrease downstream, not even in the entrance to the lower chamber, which is confirmed by eight inlets. The outlet area, or entrance to the lower chamber, is 4 × 10^−2^ mm^2^ and is established in this way to avoid unnecessary membrane overload in the suction process.

#### 2.2.2. Discharge Oriented Structural Design

The outlet of the micropump is composed of several structures identifiable in Figure 2. On each side, four inlets are placed and directed toward the pyramidal-shaped microvalves with a total area of 4 × 10^−2^ mm^2^. Their function is to rectify the flow direction through the coupled microchannels and finally, in direction to the microneedles.

In the last decade, different studies reported different techniques for the fabrication of drug delivery microsystems in patients. Nguyen et al., in 2013, reported a study of drug delivery microsystems based on Lab on Chip technology [15]. The techniques used in that study were a micromachining laser for the hollow metal microneedles [16] and deep reactive ion etching to fabricate the silicon microneedles. Sanjay et al. reported, in 2018, a microneedle used so that rat skin could be microinjected accurately with required depths (50–900 μm) and rates (up to 60 μL/min) [17].

### 2.3. TD^3^ Theoretical Calculations

In order to begin the simulation process, some input parameters had to be calculated. These parameters were established using the microneedle structure, which was validated by us, as seen in Figure 3, and in a previous study involving different simulation scenarios [7,12]. For not generating pain at the time of insertion in the patient, the chosen dimensions were wholly oriented, in addition to allowing the passage of fluid through the microneedle into the skin [12]. The length of the microneedle is that necessary for the tip to reach the segment of skin called the epidermis, ultimately crossing the layer called the stratum corneum. According to Davis et al., in 2005 [16], as well as Roxhed et al., in 2007 [18], the force to pierce the skin at this scale ranges between 0.5 N and 1.5 N, while also generating transverse forces between 0.5 N and 0.8 N that tend to bend the microneedle. Figure 4 and Figure 5 show an analysis was carried out to determine the structural behavior of the microneedles under the conditions of the skin insertion process. The axial force to be used in the analysis is 1 N, while the transverse force is 0.5 N. During these simulations, it is intended to observe the combined efforts, called Von-Misses, and the total deformation in both cases. Davis et al. employed laser micromachining for the fabrication of the hollow metal microneedles [16], and the dimensions indicated are consistent with the dimensions analyzed in this study and previous studies [12], where they concluded that drug delivery experiments using a 16-microneedle array inserted into the skin of diabetics reduced the blood glucose level by 47%.

The microneedles will be the structure wherewith the fluid is transdermally delivered. Therefore, it is highly essential to calculate the necessary variables like mass flow, volumetric flow, and fluid velocity in this first structure in order to obtain the initial, or input, parameters for the suction and discharge CFD simulations [6]. In a previous study, the time window was found to complete the dosing process and the dose-volume, which were 6 min and 0.4 mL, respectively [17,19]. The volumetric flow (*Q*) is calculated in Equation (1) and for a single needle as shown in Equation (2).

(1)Dose=0.4ml6min=6.67×10−2cm3min,

(2)Q=6.67×10−2cm3min38=1.75×10−3cm3min=2.92×10−5cm3S,

Once Q has been determined, the mass flow ($\overset{•}{m}$) will be the input parameter in simulations, which is calculated using both Q and the fluid density (*ρ*), as seen in Equation (3). The reported value for insulin [19] was 1.24 gr/cm^3^.

(3)$$\overset{•}{m}=Q\times \rho ,$$

## 3. TD^3^ Computational Fluid Dynamics (CFD) Approach

### 3.1. Microneedle CFD

The simulation is set up and solved to obtain the pressure contours that are required over the microneedle top, or inlet, to achieve the desired characteristics related to volumetric flow rate and fluid velocity. The last two parameters are compared with those theoretically calculated to confirm the correct physics development. The mesh is conformed by the microneedles inner volume, where the inlet boundary condition is set as a mass flow. The outlet boundary condition is set as a pressure outlet setting, and it has a zero magnitude, reflecting atmospheric pressure. This suggests that laminar flow will be predominant in all of the domains because we are describing a microfluidics flow. However, we calculated Reynolds’s number in order to confirm this premise, obtaining a value of 120. As shown in Figure 6, 1670 Pa is the required pressure on the microneedle top to ensure a Q magnitude of 2.92 × 10^−5^ cm^3^/s and a fluid velocity of 0.055 m/s inside the microneedle (Figure 7).

Keeping in mind that we are dealing with laminar flow, as Reynolds’s number indicated before, we obtained the parabolic velocity profiles as seen in Figure 7.

### 3.2. TD^3^ Discharge CFD

Using the results of the microneedle CFD simulation, the TD^3^ discharge simulation was designed to obtain the pressure, which has to be exerted by the membrane right into the lower chamber so the fluid could be able to achieve the desired characteristics. In the prior simulation, $\overset{•}{m}$ was set to 3.63 × 10^−5^ gr/s due to the single microneedle criteria, but this time as we are dealing with the entire array, so this value has been set to 1.39 × 10^−3^ gr/s. As shown in Figure 8, the red dotted line indicates the flow direction, which begins in the orifice controlled by the microvalve moving across the microchannel structure and ending in the microneedle tip. In order to achieve the required *Q*, the pressure in the lower chamber, which is exerted by the membrane, has to be 2560 Pa [12]. The shallow fluid compressibility allows us to ensure that the pressure generated in the upper chamber will be entirely transmitted to the lower chamber. The velocity right at the outlet in the microneedle couplings, as shown in Figure 9, does not differ too much compared with the microneedles inlet velocity shown in Figure 7.

### 3.3. TD^3^ Suction CFD

For the simulation of the suction movement of the micropump, the pumped volume (*Vp*) is ensuring that the lower chamber stroked volume stays below 30% of the dead volume (*Vo*) [9], in order not to schematize the operation of the membrane on a value that would lead it to have more significant wear. The prior fact means that the lower chamber volume is 1.2 × 10^−3^ cm^3^. Due to the total of 0.4 mL in 6 min, the time window of the entire dosing goal (Q), was 1.12 × 10^−3^ cm^3^/s.

The actuation frequency (F) was selected based on a previous study that corresponded to the volumetric flow, (6.6 Hz) [13]. The pumped volume, which the membrane will regulate, was calculated, as shown in Equation (4). The pumped volume will correspond to 1.7 × 10^−4^ cm^3^, which corresponds to 14% of the dead volume.

(4)$$Q=V_{P}\times F,$$

The inlet parameter for this simulation will be $\overset{•}{m}$, calculated with the prior *Q* and the insulin density through Equation (3), resulting in a 1.39 × 10^−3^ gr/s magnitude. With this simulation, we seek to determine the negative pressure, or suction, the membrane must exert in the lower chamber to achieve 1.12 × 10^−3^ cm^3^/s in all three micropumps. As shown in Figure 10, the red dotted line represents the fluid flow starting in the pyramid valves inlets and then flowing directly to the lower chamber due to the generated suction. The simulation results established that the negative pressure, or suction, the membrane must exert to achieve the desired flow condition is 116 Pa, in accord with values previously reported in 2015 [14], as shown in Figure 11. The inlet velocity produced by the mentioned pressure effect is 0.06 m/s.

## 4. Conclusions

A novel TD^3^ MEMS design was CAD modeled and analyzed under normal operating conditions. The conclusions are summarized below. Ensuring a 30% lower chamber stroked volume, the membrane operation frequency will not need to exceed 6.6 Hz. The micropump device is capable of dosing 2.92 × 10^−5^ cm^3^/s in a 6.6 Hz membrane pumping frequency configuration with a fluid velocity of 0.055 m/s. CFD simulations showed that during the discharge action, the membrane would have to exert 2560 Pa of pressure on the lower chamber in order to ensure a 3.63 × 10^−5^ gr/s mass flow. During the suction action, the membrane will have to exert 116 Pa of negative pressure to be able to refill the 14% stroked volume of the lower chamber at a 1.39 × 10^−3^ gr/s mass flow rate.

The discharge pressure and suction pressure values of 2.56 KPa and 116 Pa, respectively, as well as the expansion volume and dead volume, 4.2 × 10^−4^ cm^3^ and 1.2 × 10^−3^ cm^3^, respectively, will serve as a starting point for the next stage of the project-oriented design of the actuator PZT membrane.

## Figures and Tables

**Figure 1 sensors-19-05090-f001:**
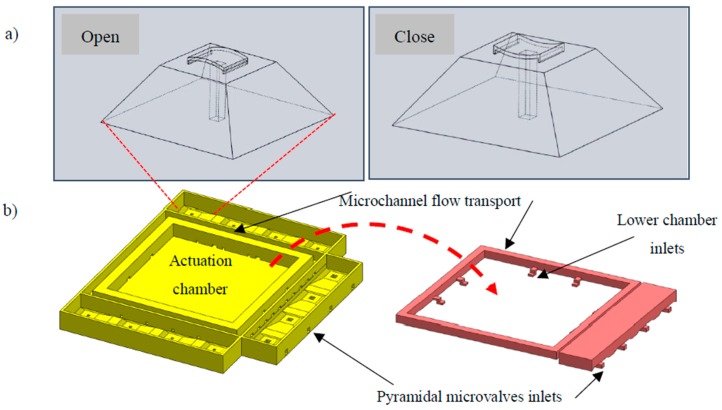
(**a**) The microvalves open and close functions, and (**b**) the fluid volume suction structure of the micropump.

**Figure 2 sensors-19-05090-f002:**
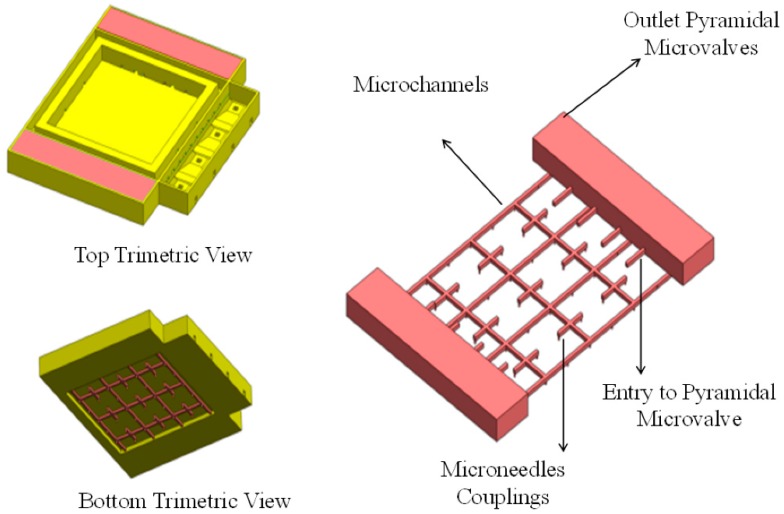
Fluid volume discharge structure.

**Figure 3 sensors-19-05090-f003:**
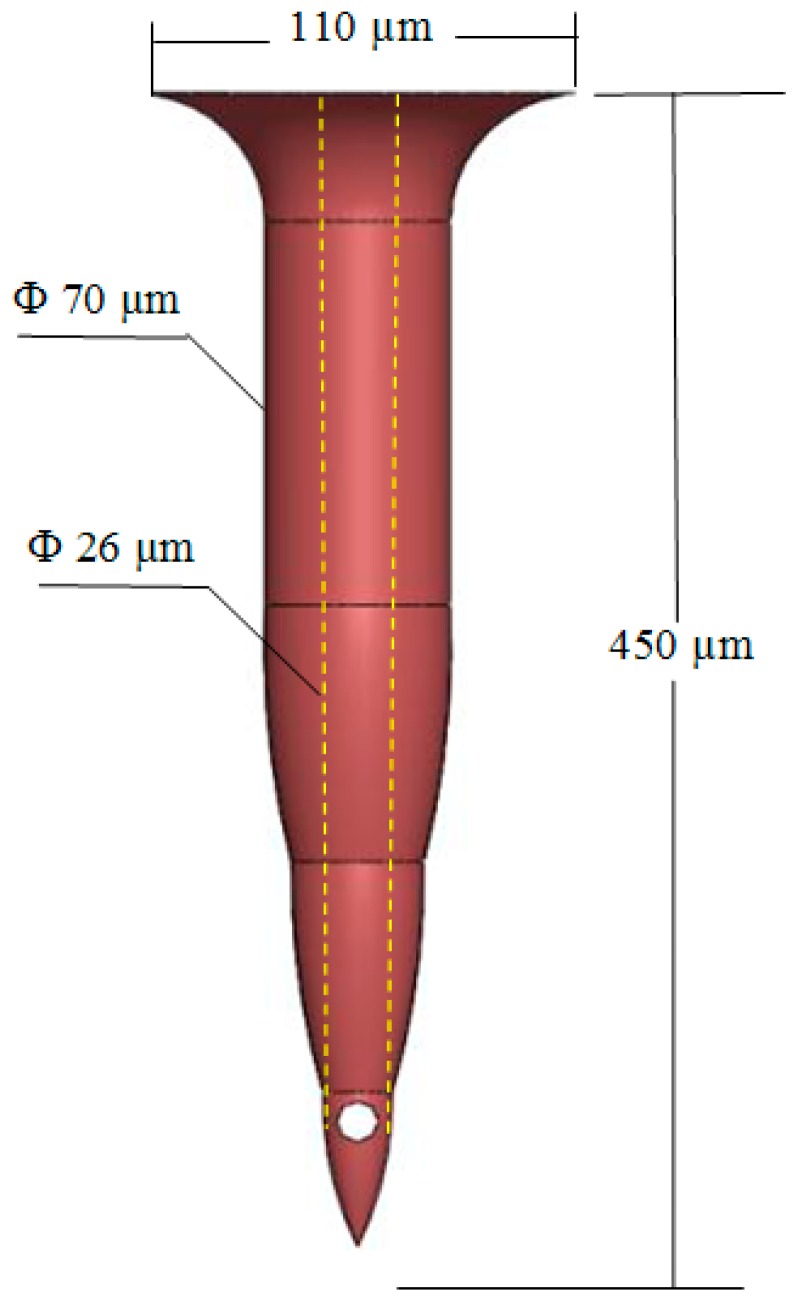
Bee sting microneedle.

**Figure 4 sensors-19-05090-f004:**
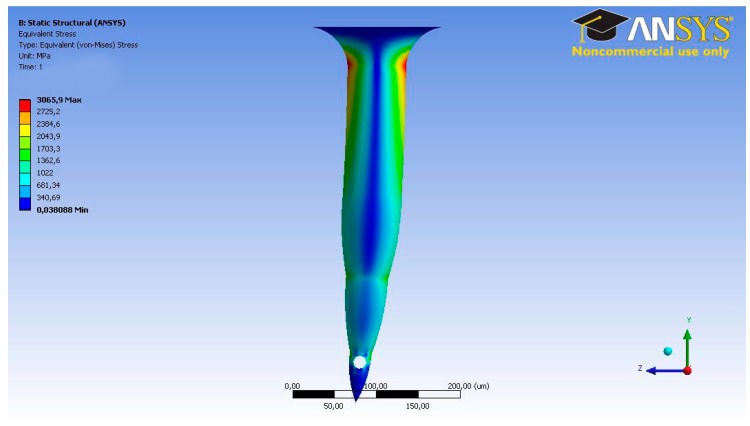
Finite element analysis (FEA) results. Von-Misses stresses under axial force for Bee’s sting microneedle.

**Figure 5 sensors-19-05090-f005:**
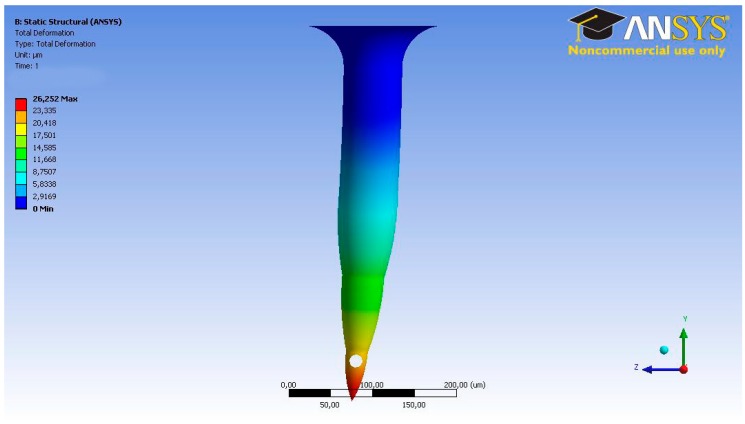
FEA results. Total deformation stresses under axial force for Bee’s sting microneedle.

**Figure 6 sensors-19-05090-f006:**
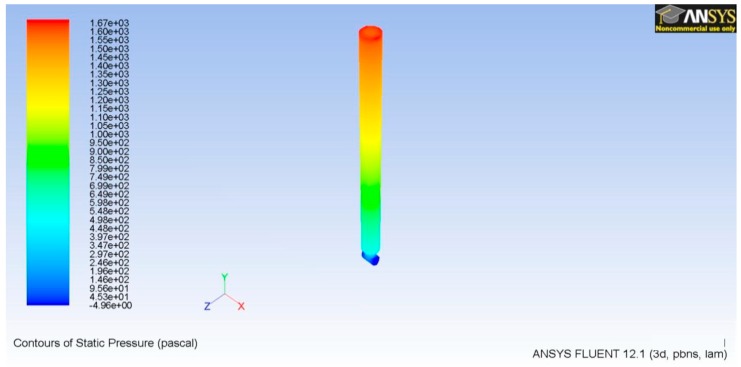
Contours of pressure for the microneedle.

**Figure 7 sensors-19-05090-f007:**
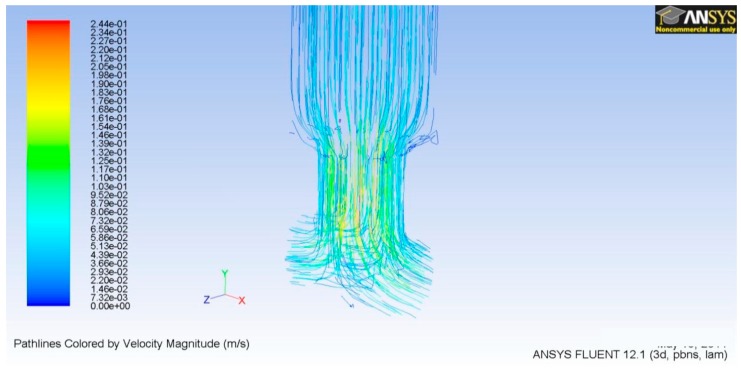
Velocity vectors for the microneedle outlet.

**Figure 8 sensors-19-05090-f008:**
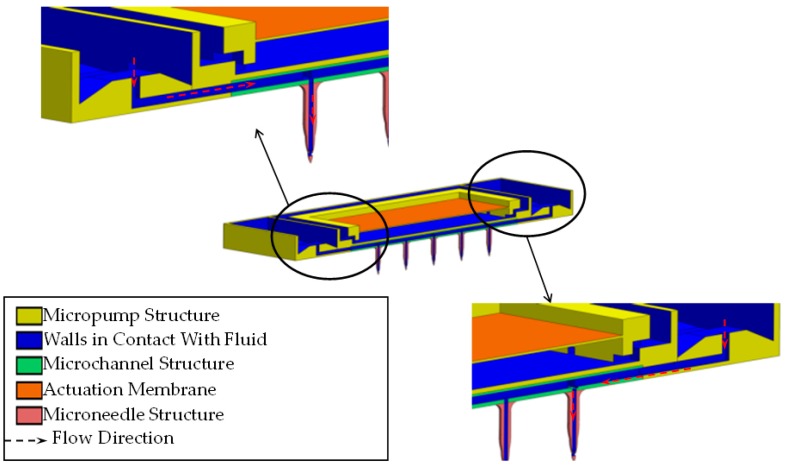
Cross sectional view of the discharge process.

**Figure 9 sensors-19-05090-f009:**
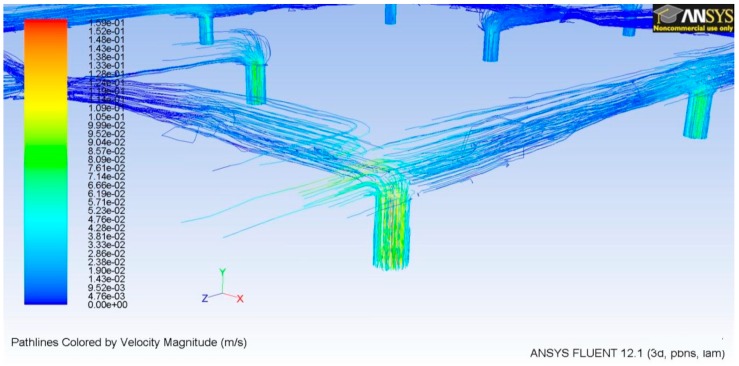
Velocity vectors of the microneedle couplings at the discharge section.

**Figure 10 sensors-19-05090-f010:**
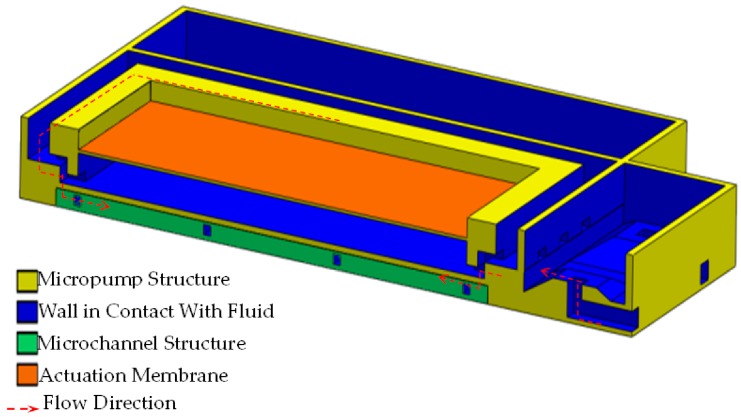
Cross sectional view of the suction process.

**Figure 11 sensors-19-05090-f011:**
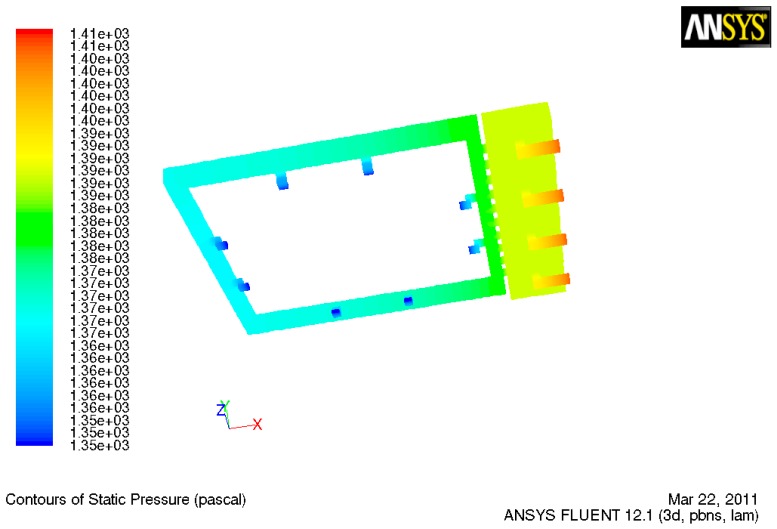
Contours of the pressure suction section.

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
