# Peer review of "Design and Analyses of a Transdermal Drug Delivery Device (TD^3^) [Author-notes fn1-sensors-19-05090]"

_sensors, 2019, doi:10.3390/s19235090_

Round 1
Reviewer 1 Report
The manuscript reports on the design and simulation of a MEMS-based transdermal drug delivery system possibly exploiting thermopneumatic or piezoelectric actuation mechanisms. Computational fluid dynamics (CFD) simulations were used to calculate the operational frequency and pressure of the membrane needed for pumping 0.4 mL of a model liquid into a microneedles array and refilling the reservoir chamber in 6 min. Transdermal drug delivery surely is a valid, painless, and minimally invasive path for medicine administration and MEMS can offer remarkable opportunities in this research area. However, the paper does not advance this field, provides only little progresses in terms of design rules, operational parameters-to-performance relationships, and implementation of the proposed device, and this certainly is not enough to warrant publication in Sensors. Here below detailed comments supporting my decision:
1) Comment: The authors claimed that actuation methods considered for the membrane are either thermopneumatic or piezoelectric ones, but they did not provide any details on their implementation and design. The only parameter investigated is the pressure that the membrane must transfer to the liquid for pumping and withdrawing. Which is the temperature or electric voltage necessary for reaching such a pressure by means of thermopneumatic or piezoelectric actuators? How the thermopneumatic or piezoelectric deflection of the membrane is implemented?
2) Comment: Calculating the pressure necessary to move a selected volume of liquid in target time is a very minimal progress. Answers to relevant questions regarding the proposed MEMS system are missing. How do geometries of the chamber, valves, needles (density, geometries), microchannels (geometries), membrane (geometries, materials, actuation mechanisms), and fluids (viscosity, density) affect the delivery performance (e.g. delivered volume, speed, pressure losses, dead volumes, liquid options) of the device?
3) Comment: I have a problem with the results obtained from simulations. The authors calculated a velocity of 0.23 m/s that for a needle with a diameter of 26 µm (from Figure 4) results into a volumetric flow rate of approximately 1.2 10^-4 cm^3/s (more than one order of magnitude of the flow rate assumed for each needle!). How is it possible?
4) Comment: In order to study the suction operations, a 30% of the volume enclosed in the lower chamber has been considered. Why? How did you choice this parameter? Criteria and rules should be in the text.
5) Comment: Initially, the authors assumed a time window to complete the dosing process of 6 min. However, the membrane frequency (F) has been calculated considering a temporal window of 0.156 minutes, which probably results from 6 min/38 # needles. Hence, we have two possible situations: either the time to complete the dosing is at least 12 min (6 min for discharge and 6 min for suction); or at F=3.1 Hz the dose is 0.2 mL that corresponds to half of the desired value (0.4 mL). In any case I would say that something is wrong in the calculation/assumption.
6) Comment: Figure 1 is useless. Figure 2-3 do not provide a clear understanding of the device architecture and its parts or components.
Comment: There are several typos and errors in English grammar. Please correct.
Author Response
Response to reviewer’s comments
The authors are very thankful to the reviewer for his/her valuable review comments to augment the quality of the submitted paper and would like to thank the publisher first for taking us into account for this special edition of Sensors. The authors have incorporated the desired answers to the comments in the manuscript:
Reviewer #1:
Comment 1: The authors claimed that actuation methods considered for the membrane are either thermopneumatic or piezoelectric ones, but they did not provide any details on their implementation and design. The only parameter investigated is the pressure that the membrane must transfer to the liquid for pumping and withdrawing. Which is the temperature or electric voltage necessary for reaching such a pressure by means of thermopneumatic or piezoelectric actuators? How the thermopneumatic or piezoelectric deflection of the membrane is implemented?
Response:
Dear reviewer, thank you very much for your comment, you are right not to introduce a study on the membrane in the final design. For this reason, is that it was not a research objective, given the fact that this design could work together with another group in the future. But given the importance of your comment, we have introduced information from a study that can be used for this presented design, remaining inside in TD 3 Actuation Principle:
“ In recent years several researchers have worked on the development of PZT piezoelectric actuators, in 2016 Dr. Shoji in Japan reported a study showing the development of a PZT actuator based on Nafion-117, wherewith a signal square-wave polarization waveform of ±2.0V, where each voltage was maintained for 2 s, finally the fabricated micropump effectively transported water at ∼ 5 L/s. In the same year, researches of the Department of Mechanical and Materials Engineering, Queen’s University from Canada, report the development of a PZT actuator based on PDMS, with a peak flow rate of 135 µL/min and a maximum back pressure of 25 mmH 2 O at an actuation frequency of 12 Hz. And in India reported other study of PZT PDMS with a pressure-flow characteristics of the micropump of maximum back pressure 220 Pa at maximum flow rate 20 µL/min. ”
Comment 2: Calculating the pressure necessary to move a selected volume of liquid in target time is a very minimal progress. Answers to relevant questions regarding the proposed MEMS system are missing. How do geometries of the chamber, valves, needles (density, geometries), microchannels (geometries), membrane (geometries, materials, actuation mechanisms), and fluids (viscosity, density) affect the delivery performance (e.g. delivered volume, speed, pressure losses, dead volumes, liquid options) of the device?
Response:
Dear reviewer, we agree with you, but as we explained earlier and we thank you for making those comments, it is a study that will be carried out in the company of other researchers.
Comment 3: I have a problem with the results obtained from simulations. The authors calculated a velocity of 0.23 m/s that for a needle with a diameter of 26 µm (from Figure 4) results into a volumetric flow rate of approximately 1.2 10^-4 cm^3/s (more than one order of magnitude of the flow rate assumed for each needle!). How is it possible?
Response:
Dear reviewer, thank you very much for your comment, you are right, and I’m sorry for the mistake. We introduce the change in the document “…As shown in Figure 4, 1670 Pa is the required pressure on the microneedle top to ensure a Q magnitude of 2,92x10-5 cm3/s and a fluid velocity of 0,055 m/s inside the microneedle (Figure 6)...”
Comment 4: In order to study the suction operations, a 30% of the volume enclosed in the lower chamber has been considered. Why? How did you choice this parameter? Criteria and rules should be in the text.
Response:
Dear reviewer, thank you very much for your comment, we introduce in the document the selection criteria of the volume in the lower chamber”… For the simulation of the suction movement of the micropump, the pumped volume (Vp), ensuring lower chamber stroked volume below to 30% of the dead volume (Vo), in order not to schematize the operation of the membrane on a value that would lead it to have more significant wear…”
Comment 5: Initially, the authors assumed a time window to complete the dosing process of 6 min. However, the membrane frequency (F) has been calculated considering a temporal window of 0.156 minutes, which probably results from 6 min/38 # needles. Hence, we have two possible situations: either the time to complete the dosing is at least 12 min (6 min for discharge and 6 min for suction); or at F=3.1 Hz the dose is 0.2 mL that corresponds to half of the desired value (0.4 mL). In any case I would say that something is wrong in the calculation/assumption.
Response:
Dear reviewer, thank you very much for your comment. The time window is a result of previous research conducted by Dr. Bao and colleagues in 1997, reference 15. But according to the design criteria we change the calculation of the actuation frequency (F) to calculate the pumped volume (Vp), according to the established volumetric flow and the actuation frequency (F) according to the PZT actuator presented by Dr. Kawun et al in 2016 [13]. Thus obtaining a pumped volume below 30% as established by the design criteria.
Comment 6: Figure 1 is useless. Figure 2-3 do not provide a clear understanding of the device architecture and its parts or components.
Response:
Dear reviewer, we have removed Figure 1 and changed Figure 2 so that it can be better understood with Figure 3.

Reviewer 2 Report
In this paper the authors introduced the design of a TD3 MEMS device for transdermal drug delivery system using computational simulations.
The authors indicate that is a novel type of device however, there is no explanation concerning what is the improvement compared to the existing devices.
The main problem of this paper is the absence of the discussion and the comparison with the other research works. The authors need to explain what improvements their device has over existing ones. The introduction need to be completed by the actual state of the art in this field, the recent reviews as for example HyunjaeLee et alhttps://doi.org/10.1016/j.addr.2017.08.009 could allow to have actualized data.
In Line 16 : “The TD3 has shown to be capable of delivering a volumetric 16 flow of 2,92x10-5 cm3/s with a 3,11 Hz membrane stroke frequency”. There are not experimental data in this paper, then the authors calculated an hypothetical behavior but not the real capacity of delivery.
References must to be completed and actualized.
Some terms must to be defined ie CAD techniques.
Author Response
Comment 1: In this paper the authors introduced the design of a TD3 MEMS device for transdermal drug delivery system using computational simulations. The authors indicate that is a novel type of device however, there is no explanation concerning what is the improvement compared to the existing devices.
Response:
Dear reviewer, thank you very much for your comment. The novelty of the design is to be able to introduce a new microneedle arrangement design to achieve insulin dosing according to previous research and the analysis with a micropump design. We introduce in the document the following text, “…A previous study was presented on the design of the microneedle to understand the best structure that allows delivering the necessary dosage for a patient, without damage the insulin molecule or the patient can feel pain [12]. The submitted contribution of the microsystem design is in accord with the result of several researchers [3,11,13], which will allow full integration in the future in the third generation transdermal drug delivery [3]...”
Comment 2: The main problem of this paper is the absence of the discussion and the comparison with the other research works. The authors need to explain what improvements their device has over existing ones. The introduction need to be completed by the actual state of the art in this field, the recent reviews as for example HyunjaeLee et alhttps://doi.org/10.1016/j.addr.2017.08.009 could allow to have actualized data.
Response:
Dear reviewer, thank you very much for your comment, you are right, and I’m sorry for the mistake. We introduce various changes in the document with four new references, including your suggestion, as like reference 3. We use the reported results to compare the results obtained in this study.
Comment 3: In Line 16 : “The TD3 has shown to be capable of delivering a volumetric 16 flow of 2,92x10-5 cm3/s with a 3,11 Hz membrane stroke frequency”. There are not experimental data in this paper, then the authors calculated an hypothetical behavior but not the real capacity of delivery.
Response:
Dear reviewer, yes, it’s true, we present results calculated by means of simulations that allow obtaining hypothetical behavior, not the real behavior of the device's operation. But we compared the results obtained with the result reported for various researches
Comment 4: References must to be completed and actualized.
Response:
Dear reviewer, thank you very much for your comment. We introduce four new references and updates, including your suggestion, as like reference 3.
Comment 5: Some terms must to be defined ie CAD techniques.
Response:
Dear reviewer, thank you very much for your suggestion. We introduce the main of CAD techniques.

Round 2
Reviewer 1 Report
The authors didn’t address (not completely) the issues I highlighted during the revision process. Regarding my first comment, I can understand that the integration of the proposed system with thermo-pneumatic or piezoelectric actuators represents a future prospective of the authors. However in the present form the paper provides very little advances to the field. The actuators are missing, the device doesn’t exist, potential fabrication steps to translate the device in practice ignored. How does this paper advance the field of the drug delivery? Why people should read or be interested to this paper? As for my second comment, determining the structure-to-property relationships of the device (for instance, the delivered volume and speed as a function of the geometries of the chamber, valves, needles, microchannels, and membrane) requires additional simulations that, I humbly believe, the authors could implement without waiting for future collaborations with other groups. I regret to tell you that the paper, though partially strengthened, still has serious problems and its scientific content is not high enough to warrant publication in Sensors.
Author Response
Comment 1: The authors didn’t address (not completely) the issues I highlighted during the revision process. Regarding my first comment, I can understand that the integration of the proposed system with thermo-pneumatic or piezoelectric actuators represents a future prospective of the authors. However in the present form the paper provides very little advances to the field. The actuators are missing, the device doesn’t exist, potential fabrication steps to translate the device in practice ignored. How does this paper advance the field of the drug delivery? Why people should read or be interested to this paper?.
Response:
The highlight is to be able to show structural analysis of the delivery needed for a patient with diabetes, including results of the structure of the microneedles coupled with microchannels for a transdermal system. The study provides computational simulation results that are consistent with results reported by other authors separately. It allows us having a design that can be fabricated with techniques used for the development of third-generation transdermal drug delivery.
Comment 8: determining the structure-to-property relationships of the device (for instance, the delivered volume and speed as a function of the geometries of the chamber, valves, needles, microchannels, and membrane) requires additional simulations that, I humbly believe, the authors could implement without waiting for future collaborations with other groups. I regret to tell you that the paper, though partially strengthened, still has serious problems and its scientific content is not high enough to warrant publication in Sensors.
Response:
Dear reviewer, we have introduced new reports showing several microfabrication techniques that would allow the development of the designed TD3 in the future and allow us to compare the results of our simulations with the results reported in the last decade. We introduced new simulations and structural analysis of the microneedles to improving the study, to enhance the support of the research presented.

Reviewer 2 Report
Accept in present form
Author Response
Dear reviewer, thank you very much for your comments and approval.
